# The Use of Distinctive Monoclonal Antibodies in FMD VLP- and P1-Based Blocking ELISA for the Seromonitoring of Vaccinated Swine

**DOI:** 10.3390/ijms23158542

**Published:** 2022-08-01

**Authors:** Heng-Wei Lee, Cheng-Yao Yang, Ming-Chang Lee, Shih-Ping Chen, Hui-Wen Chang, Ivan-Chen Cheng

**Affiliations:** 1School of Veterinary Medicine, National Taiwan University, Taipei 106, Taiwan; d06629010@ntu.edu.tw (H.-W.L.); huiwenchang@ntu.edu.tw (H.-W.C.); 2Graduate Institute of Veterinary Pathobiology, National Chung Hsing University, Taichung 402, Taiwan; yangchengyao@nchu.edu.tw; 3Agricultural Technology Research Institute, Hsinchu 300, Taiwan; lee@mail.atri.org.tw (M.-C.L.); spchen@mail.atri.org.tw (S.-P.C.); 4Graduate Institute of Molecular and Comparative Pathobiology, School of Veterinary Medicine, National Taiwan University, Taipei 106, Taiwan

**Keywords:** foot-and-mouth disease virus, virus-like particles, blocking ELISA, monoclonal antibodies

## Abstract

The serum neutralization (SN) test has been regarded as the “gold standard” for seroconversion following foot-and-mouth disease virus (FMDV) vaccination, although a high-level biosafety laboratory is necessary. ELISA is one alternative, and its format is constantly being improved. For instance, standard polyclonal antisera have been replaced by monoclonal antibodies (MAbs) for catching and detecting antibodies, and inactive viruses have been replaced by virus-like particles (VLPs). To the best of current knowledge, however, no researchers have evaluated the performances of different MAbs as tracers. In previous studies, we successfully identified site 1 and site 2 MAbs Q10E and P11A. In this study, following the established screening platform, the VLPs of putative escape mutants from sites 1 to 5 were expressed and used to demonstrate that S11B is a site 3 MAb. Additionally, the vulnerability of VLPs prompted us to assess another diagnostic antigen: unprocessed polyprotein P1. Therefore, we established and evaluated the performance of blocking ELISA (bELISA) systems based on VLPs and P1, pairing them with Q10E, P11A, S11B, and the non-neutralizing TSG MAb as tracers. The results indicated that the VLP paired with S11B demonstrated the highest correlation with the SN titers (R^2^ = 0.8071, *n* = 63). Excluding weakly positive serum samples (SN = 16–32, *n* = 14), the sensitivity and specificity were 95.65% and 96.15% (kappa = 0.92), respectively. Additionally, the P1 pairing with Q10E also demonstrated a high correlation (R^2^ = 0.768). We also discovered that these four antibodies had steric effects on one another to varying degrees, despite recognizing distinct antigenic sites. This finding indicated that MAbs as tracers could not accurately detect specific antibodies, possibly because MAbs are bulky compared to a protomeric unit. However, our results still provide convincing support for the application of two pairs of bELISA systems: VLP:S11B-HRP and P1:Q10E-HRP.

## 1. Introduction

Foot-and-mouth disease virus (FMDV) is highly contagious and affects all susceptible cloven-hoofed animals globally. In addition to the lack of cross-protection among the seven serotypes (O, A, C, Asia1, SAT1, SAT2, and SAT3), restrictions on the import of animal products are very strict, particularly those in FMDV-free countries. A facility with a high biosafety level is also required to handle live FMDV, which carries the potential risk of viral leakage. An effective countermeasure against FMDV outbreaks is blanket vaccination with selected strains followed by serum neutralization (SN) tests to determine the protective antibody response [1,2]. However, both inactivated vaccine production and the SN test are highly restricted to high-level biosafety laboratories.

In 1997, a devastating FMDV outbreak resulted in tremendous economic loss in Taiwan. The virus strain, O/TW/97 (O/97), was characterized by porcinophilic activity [3]. Following the guidelines of the OIE (Office International des Epizooties/World Organisation for Animal Health), mandatory vaccination with inactivated vaccines made using three strains—O 4174, O1 Campos, and O1 Manisa—was recommended by the World Reference Laboratory based on an evaluation of r1 values greater than 0.3 [4] immediately following the outbreak. The FMDV strain O/98, a homologue of O/97, was produced and recognized as the vaccine strain later in Taiwan. After blanket vaccination and surveillance, O/Penghu/2012 was the last isolated virus from this series of FMD cases. In 1999, another virus strain was isolated in Kinmen, an archipelago far off the western coast of Taiwan. It was named O/TW/2/99 and classified as the PanAsia topotype, which is separate from Cathay O/TW/97 [5,6]. The FMD eradication campaign lasted until 2020, when Taiwan was recognized as an FMDV-free country by the OIE.

FMDV is non-enveloped and consists of a single-stranded positive-sense RNA genome surrounded by a densely packed icosahedral protein shell. The shell comprises 60 copies of a protomer with four capsid proteins, VP1 to VP4, where VP4 is located on the inner side of the virion. FMDV targets host cell receptors, including integrins (αvβ1, αvβ3, αvβ6, and αvβ8), heparan sulfate, and JMJD6, for viral entry [7,8]. However, protective antibodies could effectively block the interaction between virions and receptors to neutralize the virus. Based on MAb escape mutant studies, five neutralizing sites have been described for serotype O. Site 1, located at the GH loop and carboxy terminus of VP1, includes VP1–144, 148, 150, and 208. The critical residues of site 2 are VP2–70, 71, 72, 73, 75, 77, 131, and 191. The residues in site 3 include VP1–43 and 44 at the BC loop of VP1 near the fivefold axis. Site 4 is located at VP3 (VP3–56 and 58), and site 5 at the GH loop of VP1 (VP1–149) [9,10,11,12,13]. Although these findings provide a basic understanding of neutralizing antibodies, the neutralizing sites differ among serotypes [14]. Additionally, with the exception of site 1, the neutralizing sites have conformational structures and cannot be presented by synthetic peptides [9]. Therefore, a platform based on virus-like particles (VLPs) established in a previous study was used to successfully identify more than six site 2 MAbs whose binding to VLPs could be abrogated or inhibited by VP2-S72N mutation on the VLPs [15]. The platform preserved the authentic antigenicity without using live or inactivated viruses. We presumed this could also apply to the screening and characterization of all neutralizing monoclonal antibodies.

The SN test is prescribed as a standard method for vaccine seromonitoring, even though it is laborious, expensive, and time-consuming, and it requires a high-level biosafety laboratory. In addition to the biological variability in SN tests, ELISA has been adopted by many laboratories to replace the SN test, and a more advanced format for ELISA has been developed [16]. In the beginning, the solid-phase competition ELISA (SPCE) format used polyclonal antisera as capture antibodies and tracers, with inactivated, purified viruses as antigens [17]. Later on, polyclonal antibodies were replaced by MAbs as the tracer and capture for large-scale serology [18], and VLPs replaced inactivated viruses, whose manufacture was limited to BSL3 laboratories [18,19]. A recombinant VP2 subunit protein from the Egyptian SAT2 isolate and a P1 capsid polyprotein of serotype O were also chosen as the diagnostic antigens [20,21]. The rP1-SPCE system paired guinea pig anti-serum as the tracer and demonstrated excellent agreement with in-house liquid-phase blocking ELISA [21]. In serotype O, the subunit protein could not represent the conformational neutralizing sites apart from the linear epitope (site 1), which was the only exception. Therefore, our study aimed to develop a blocking ELISA system based on VLPs or P1 combined with different tracer MAbs and evaluate the most suitable pairs.

## 2. Results

### 2.1. Identification of Binding Site for S11B Monoclonal Antibody

Previously, we determined that the Q10E MAb recognized site 1 based on a synthetic GH peptide on VP1 [22] and successfully proved the P11A MAb as the site 2 antibody using mutated VLP/VP2-S72N [15]. Meanwhile, the S11B MAb, a neutralizing antibody, was found not to recognize the unprocessed protomer P1. After processing and assembly, the capsid proteins formed a higher-order oligomer, possibly a pentamer, which could be bound to S11B [15]. To examine the binding of S11B at the neutralizing site, we expressed VLPs from putative escape mutants of O/TW/97, which included the VP1-L148R mutant for site 1, VP2-S72N and VP2-S131P mutants for site 2, VP1-P44L and VP1-K43A/P44A for site 3, VP3-D58K and VP3-H56A/D58A for site 4, and VP1-Q149H for site 5 (Figure 1A). As expected, the mutation at site 1 (VP1-L148R) almost abrogated the binding of the Q10E MAb (Figure 1B). The mutation of VP2-S72N inhibited the interaction between P11A and VLPs, while VP2-S131P had no significant impact (Figure 1C). Because the double mutations at site 3 (VP1-K43A/P44A) significantly inhibited the binding of S11B, we presumed that S11B was the site 3 MAb (Figure 1D). For normalizing the VLPs, the OD values of the indicated VLPs were adjusted using those from Q10E, except for the site 1 mutated VLP, and the same conclusion was reached (Appendix A). The same was true when the values were adjusted using those from P11A, except for the site 2 mutated VLP (Appendix A). In short, we produced and identified at least three neutralizing MAbs: Q10E at site 1, P11A at site 2, and S11B at site 3, as indicated in Table 1.

### 2.2. Establishment of VLP- and GST-P1-Based bELISA

The three neutralizing MAbs (Q10E, P11A, and S11B) and the non-neutralizing MAb TSG were applied as tracers in blocking ELISA paired with VLPs and P1. Excluding the S11B-HRP and P1 pair, seven pairs were tested using a small number (*n* = 6) of samples from FMDV vaccinated swine, including SN512, SN256-1, SN256-2, SN64, SN16-1, and SN16-2. Meanwhile, to assess the specificity, negative sera (*n* = 12) against FMDV were also tested, including sera from five specific pathogen-free (SPF) pigs; sera against pseudorabies (PR), classic swine fever (CSF), and swine vesicular disease (SVD); in addition to three negative sera from imported pigs. The PI values of the positive samples were above 40% in the VLP pairs with Q10E-HRP, TSG-HRP, or S11B-HRP. The negative samples had values under 20% (Figure 2). The combination of the VLP and S11B-HRP performed particularly well; the PI values were 80–95% for the positive antisera and lower than 16% for the negative sera (Figure 2).

### 2.3. Evaluation of VLP- and GST-P1-Based bELISA

To select a suitable pair for bELISA that was positively correlated with the SN titers, we applied 63 samples, including 10 SPF sera (Appendix A). According to the OIE manual, samples with SN titers below 16 are considered negative. For positive anti-FMDV sera, the titer should exceed 16, with SN titers of 16–32 indicating a weak positive [4]. Therefore, we used strongly positive sera from vaccinated pigs (*n* = 23), weakly negative sera (*n* = 14), and negative sera (*n* = 26). The means of triplicate samples were calculated and are demonstrated in Figure 3 and Appendix A. The PI values for all pairs demonstrated a statistically significant correlation with those of the SN test (*p* < 0.0001); however, the VLP paired with S11B-HRP appeared to be the best combination (R^2^ = 0.8071), better than the P1 and Q10E-HRP pairing (R^2^ = 0.7680). The optimal cut-off value for each blocking ELISA was analyzed and determined using ROC analysis (the weak, doubtful positive samples were excluded) (Appendix A). The sensitivity and specificity values were 95.65 and 96.15%, respectively, for the VLP paired with S11B-HRP, indicating very high agreement (kappa = 0.92) (Table 2). Although the P1 and Q10E-HRP pair exhibited relatively low sensitivity (78.26%), the SN titers of five false-negative samples ranged from 64- to 128-fold, while the PI values ranged from 36 to 47%, near the cut-off value of 47.45 for this pair (Appendix A).

### 2.4. Steric Effect between Q10E, P11A, S11B, and the Non-Neutralizing Antibody TSG

Despite the distinct binding epitopes of the four MAbs, we discovered that they interfered with each other within the same antigens (Figure 4A). To verify this, non-conjugates were applied in serial two-fold dilutions (25–200 ng/well) to block the VLPs, followed by the indicated tracers (Figure 4B). The highest PI values in each group were observed for the pairs with identical MAbs, such as Q10E blocking Q10E-HRP (Figure 4A). However, Q10E-HRP was also blocked by P11A and S11B, which were individually directed to site 2 and site 3, respectively. In contrast, the degree of blocking for P11A-HRP by Q10E was not evident. This trend agreed with the results of the P1-based bELISA (Appendix A). The PI values in the TSG-HRP group, ranked from high to low, were those of the pairs with TSG, P11A, S11B, and Q10E. S11B-HRP was also blocked by S11B, Q10E, and P11A (Figure 4B).

To exclude the possibility that the tracers replaced the blocking antibodies, we labeled the blocking antibodies by reversing the order of addition of non-conjugates and conjugates. The extremely low PI values indicated that the conjugates could still bind to the VLP, even after the addition of 400 ng of non-conjugates (Appendix A). Therefore, in Figure 4B and Appendix A, the TSG bound to the VLP and P1 was not replaced by P11A-HRP, implying that P11A-HRP and TSG coexisted on the antigens. However, TSG-HRP was blocked by P11A on the VLP and P1 (Figure 4B and Appendix A). We presumed that the affinity of the tracers (added later) determined the ability to coexist on an antigen with blocking antibodies. Therefore, the PI values reflect not only the binding sites but also the affinity of the indicated tracers when testing the same subjects (blocking antibodies).

## 3. Discussion

Scientists are currently attempting to reform the format of blocking ELISA to detect protective antibodies against FMDV [16,17,18,19,23,24]. However, to our knowledge, no comparative studies of different tracers and different antigens using VLPs and P1 have been reported. Our study identified S11B as the site 3 MAb based on our established platform. The other MAbs, Q10E, P11A, and TSG, were chosen as tracers to represent site 1, site 2, and non-neutralizing antibodies, respectively. Excluding the S11B and P1 pair, the performance of the seven pairs was compared with 63 swine sera, of which 53 samples were collected from vaccinated pigs and 10 from SPF pigs. VLP:S11B-HRP and P1:Q10E-HRP indicated high performance in detecting protective antibodies and were positively correlated with SN titers (R^2^ = 0.8071 and 0.7680, respectively).

Given the highly folded nature of FMDV capsids and their dense packing to form virions, antigenicity should be given serious consideration in vaccines and ELISA detection systems. In the case of serotype O, for example, apart from linear site 1, the other four neutralizing sites are conformational epitopes, which cannot be represented by single subunit proteins [9,25]. For example, S11B, a site 3 MAb, could not recognize VP1 (data not provided) or the unprocessed P1 protomer [15], indicating that site 3 was formed only after the processing and assembly of P1. Similarly, Dong et al. highlighted the subtle differences in the structures of a pentamer and a virion using cryo-EM, indicating that the virion package caused a structural shift in the BC and EF loops of VP2 [26]. Moreover, Wang et al. found that the native site 1 of the serotype Asia 1 may be conformational in structure [27]. Conversely, all site 2 MAbs in our laboratory failed to recognize recombinant VP2 [15].

Although we obtained different R^2^ values from the seven pairs for bELISA, the PI values did not represent the amounts of the tested neutralizing antibodies when the indicated site MAb was used as a tracer. Due to the bulk of antibodies being compared to a pentameric unit and the repeating units of VLPs, different MAbs could compete for association with the VLPs, despite their binding to distinct epitopes. To prove this, we chose non-conjugated MAbs to represent the indicated site antibodies in sera. Site 1 was positioned close to site 2; therefore, P11A significantly blocked Q10E-HRP. Additionally, Q10E-HRP was also blocked by S11B, a site 3 MAb. TSG-HRP was blocked by P11A and S11B, implying that portions of the site 2 and site 3 antibodies were detected when the non-neutralizing TSG MAb was used as a tracer. Furthermore, we discovered that the blocking effects were not reciprocal. For instance, P11A blocked TSG-HRP, but TSG did not block P11A-HRP. We also discovered that the blocking antibodies stayed on the antigen, predominantly based on the fact that the exchange of non-conjugates and conjugates led to a PI value near zero. The predominance of the order of addition excluded the possibility that P11A-HRP replaced TSG MAb, which was already bound to the VLP or P1. We presume that the affinity of the tracer may explain why the blocking effects were not reciprocal, as described in more detail below. Given that the viruses were mixed with samples prior to contacting the host receptors during the SN test procedure, we decided solid-phase blocking ELISA, which is step-by-step, would be more reasonable than competitive ELISA, in which tracers and serum antibodies compete for the epitope at the same time.

Based on the results demonstrated in Figure 4, we defined the region of steric effect into “core” and “marginal” regions. The core region was the exact binding site. If the core regions of antibodies overlapped, the antibodies could not coexist on the same protomeric unit. Conversely, the marginal region was blocked by the Fc portion or the other hanging Fab of the blocking antibodies. A high tracer affinity may have been resistant to the steric effect in the marginal regions, resulting in low PI values. The opposite would result in high PI values, such as those observed for TSG-HRP with P11A. Despite the effect of tracer affinity, it has been reported that some neutralizing antibodies not only block viruses but also mimic receptors, resulting in viral uncoating [26]. When two Fabs of antibodies simultaneously bind to adjacent protomers, they mask a large area of the virion [10]. These two types of antibodies within samples could lead to the misinterpretation of PI values. Theoretically, soluble viral receptors [28] might be ideal tracers for bELISA. Nonetheless, there are at least three FMD viral receptors: integrins, heparan sulfate, and JMJD6. Considering their application and our results, purified MAbs were suitable for the role of tracer, as they reflected the SN titer. Moreover, non-neutralizing TSG when used as a tracer was able to detect site 2-directed antibodies.

Empty capsids, albeit with native antigenicity, are unstable and prone to dissociation [29]. There is no guarantee that VLPs will not dissociate after sucrose gradient purification, which would complicate any application. The small size and simple structure of P1 without the marginal region that VLPs carry do not induce the steric effect, which interferes with the correlation between the SN titer and data generated by bELISA. Although P1 demonstrates a slightly different virion antigenicity, it can at least present the near-native structures of site 1 and site 2 [15], both of which are immunodominant for vaccinated animals [30]. Therefore, unprocessed P1 protomers, given the promising diagnostics, are worth trying. Our results indicated no significant difference between P1 and VLPs. Although antigenicity is important for FMDV vaccines and detection systems, P1 is still a potential target for future applications due to its low production costs.

In conclusion, we established bELISA systems with VLPs and P1 and paired them with different MAbs as tracers. The systems were positively correlated with SN titers and obtained a high sensitivity and specificity (excluding weakly positive sera). Most importantly, two bELISA pairs, VLP:S11B-HRP and P1:Q10E-HRP, showed the highest performance among the seven and could be applied to large-scale serological testing and simultaneously compared with the SN test to further validate their efficacy in identifying neutralizing antibodies in swine sera samples. Additionally, sera samples from different FMD-susceptible species, such as bovines and ovines, are needed for future testing to expand and evaluate the potential for application in FMD disease surveillance.

## 4. Materials and Methods

### 4.1. Viruses and Cells

Human TK^−^ 143 (HTK^−^) cells for the recombinant vaccinia virus expression system were cultured in Dulbecco’s modified Eagle medium (DMEM, Thermo Scientific, Carlsbad, CA, USA) containing 10% Hyclone fetal calf serum (FBS). The recombinant vaccinia virus used in this study, vTF7-3, was amplified in the human TK^−^ 143 (HTK^−^) cells. The BHK-21 cells used for serum neutralization tests were maintained in DMEM containing 5% FBS.

### 4.2. Swine Sera

Swine vesicular disease virus anti-serum (SVD+ and SVD+++) (PrioCHECK SVDV Ab ELISA, 7610211 and 7610212) and FMDV-free samples after import testing (#68-1, #68-2, and #68-3) were supplied by the Animal Health Research Institute. Other FMDV-negative serum samples were supplied by the Agricultural Technology Research Institute, including sera against pseudorabies serum (PR+) (IDEXX PR-gpl Antibody Test Kit) and classic swine fever serum (CSF+) (IDEXX Classical Swine Fever Virus Antibody Test Kit) and sera from specific pathogen-free (SPF) pigs (*n* = 15). Sera from vaccinated pigs (*n* = 49) were collected from farms in the Hog Cholera and FMD Eradication Program of the Council of Agriculture, Taiwan, and supplied by the Agricultural Technology Research Institute. Serum neutralization tests for all samples were carried out using a standard SN test at the Animal Health Research Institute, Taiwan.

### 4.3. Plasmid Constructions

Point-mutated pcDNA-P1(O97) plasmids were generated using GeneArt site-direct mutagenesis (Invitrogen, Carlsabad, CA, USA, A13282) as described in a previous study [15]; the primers used are listed in Appendix A. The P1 gene was further subcloned to pET-42a to generate pET-GST-P1(O97) plasmids using *Bam*HI and *Xho*I.

### 4.4. Purification and Conjugation of MAbs

MAbs from ascetic fluid were purified with protein G (Cytiva, Marlborough, MA, USA, GE17-0618-01) and conjugated with horseradish peroxidase (HRP) (Novus Biologicals, Centennial, CO, USA, 701-0000).

### 4.5. Expression of VLPs and mVLPs for ELISA

The antigens, including VLPs and mVLPs, were produced with a recombinant vaccinia virus vTF7-3 expression system as described in previous studies [15,31]. The HTK^−^ cells were seeded in 6-well plates and infected by vTF7-3 for 1 h. The indicated plasmids containing the T7 promoter were transfected into the cells using Turbofect as per the manufacturer’s instructions (Thermo Scientific, Carlsbad, CA, USA). The molar ratio of P1 (or mutated P1) to 3C plasmids (pcDNA-3C) was 40:1 to minimize the excessive production of 3C^pro^, which is highly cytotoxic. At 20 h post transfection, the cells were harvested in 500 µL of PBS per well, followed by three freeze–thaw cycles. Cell debris was removed by centrifugation at 10,000× *g* for 20 min.

### 4.6. Sandwich ELISA

ELISA plates were coated with 400 ng/well purified TSG MAb in carbonate–bicarbonate buffer (pH 9.6) (Sigma, St. Louis, MO, USA, C3041) and held at 37 °C for 1 h. TSG was used to capture VLPs (or mutated VLPs) for antibody characterization because its binding epitope was excluded from the neutralizing sites. The plates were washed three times with PBST (PBS with 0.05% Tween-20, pH 7.3) followed by blocking with 0.5% skim milk (Sigma, 70166) in PBS for 30 min. After the removal of the blocking buffer, 1 µL of cell lysates containing VLPs (or mutated VLPs) was added at a dilution of 1:100 and incubated at 37 °C for 1 h. After washing three times, the conjugates (100 ng/well) were incubated for 1 h. After the final washing, freshly prepared TMB substrate (BD biosciences, Franklin Lakes, NJ, USA, 555214) was added and allowed to stand for 15 min. The reaction was stopped with 1 N HCl and measured at 450 nm with a Sunrise^TM^ Absorbance Reader (TECAN, Männedorf, Switzerland). Each test was performed in triplicate.

### 4.7. Expression and Purification of GST-P1

The pET-GST-P1(O97) plasmid was transferred to *Escherichia coli* (BL21) to express the recombinant GST-P1. The recombinant GST-P1 was purified from the total lysate of transformed *Escherichia coli* (BL21) by nickel-charged affinity resin (Ni Sepharose High Performance, Cytiva, Marlborough, MA, USA). The impurities were removed using phosphate-based buffer with 30 mM imidazole (Sigma-Aldrich, St. Louis, MO, USA). The recombinant GST-P1 was eluted using phosphate-based buffer with 250 mM imidazole. The imidazole was removed with a PD-10 desalting column (Cytiva, Marlborough, MA, USA).

### 4.8. Serum Neutralization Test (SNT)

The SNT was performed according to the OIE manual [32]. A two-fold serial dilution of sample (from 4×–512×) was mixed with 100 TCID_50_ of O/TAW/97 and held for 1 h at 37 °C. The suspended BHK-21 was added for infection and incubated for two days. After examination under a microscope for cytoplasmic effects, the highest antibody dilution that could still neutralize the virus at the 50% endpoint was regarded as the serum neutralization titer.

### 4.9. VLP-Based bELISA

ELISA plates were coated with 400 ng/well Q10E (or S11B) MAb and held for 1 h at 37 °C, followed by the addition of blocking buffer for 30 min. Q10E was used in standard bELISA, and S11B was used for the steric effect experiments. As a protocol for sandwich ELISA, 1 µL of VLP cell lysates were added and the mixture was held for 1 h at 37 °C. Ten-fold dilutions of each serum test sample (or dilutions of non-conjugated MAbs) in blocking buffer were added to the plates for 1 h. After washing, tracer (100 ng/well) was added and incubated for 1 h at 37 °C. After the final washing, the reaction was started using TMB substrate (BD OptEIA^TM^, San Diego, CA, USA) and stopped with 1 N HCl. The OD value at 450 nm was measured using a Sunrise^TM^ Absorbance Reader (TECAN, Switzerland). Groups with blank blocking buffer without serum samples or antibodies were used as negative controls. The percentage of inhibition (PI) for each test sample was calculated with the following formula. Each test was performed in triplicate.
$$PI\;\left(\%\right)=\left(\frac{OD.\;negative\;group-OD.\;test\;sample}{OD.\;negative\;group}\right)\times 100$$

### 4.10. P1-Based bELISA

ELISA plates were coated with anti-GST antibodies (Sigma, G7781) and held for 1 h at 37 °C, followed by the addition of blocking buffer for 30 min. Purified GST-P1 (400 ng/well) was incubated for 1 h at 37 °C. The subsequent procedures were performed as described above.

## Figures and Tables

**Figure 1 ijms-23-08542-f001:**
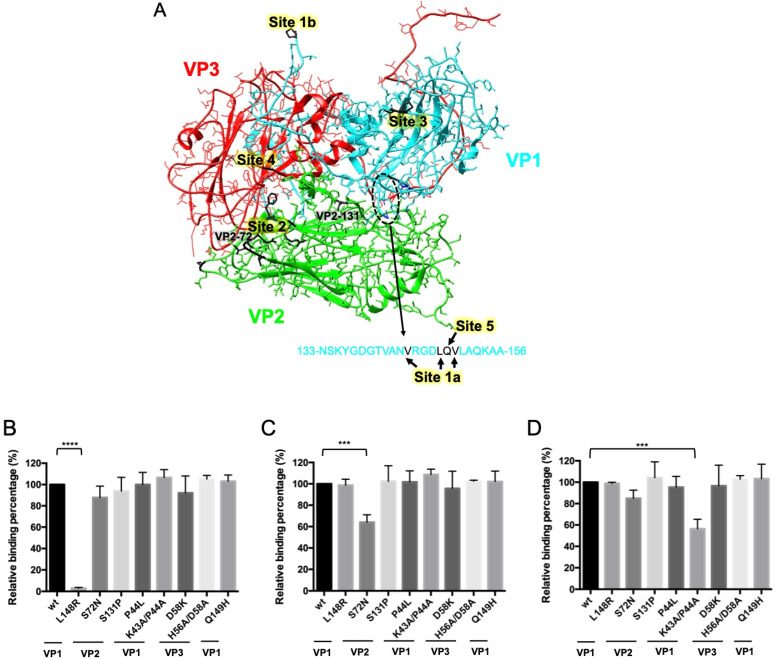
Schematic of neutralizing sites for serotype O and the relative binding percentages between MAbs and mutated VLPs. (**A**) The critical residues of the neutralizing sites were illustrated on a protomer of O1M-S093Y (PDB 5ddj) using the UCSF Chimera software. (**B**–**D**) Sandwich ELISA was performed with the VLPs and mutated VLPs to determine the relative binding strength (compared with the wild type) of the MAbs, including Q10E (**B**), P11A (**C**), and S11B (**D**). The relative binding percentages of the mutated VLPs were compared with those of the wild-type (100%) using ANOVA with Dunnett’s test. Each test was performed in triplicate. L148R—VP1-L148 mutated VLP; S72N and S131P—mutations of VLP2 in indicated amino acids; P44L and K43A/P44A—site 3 mutated VLPs; D58K and H56A/D58A—site 4 mutated VLPs; Q149H—site 5 mutated VLP. **** *p* < 0.0001, *** *p* < 0.001.

**Figure 2 ijms-23-08542-f002:**
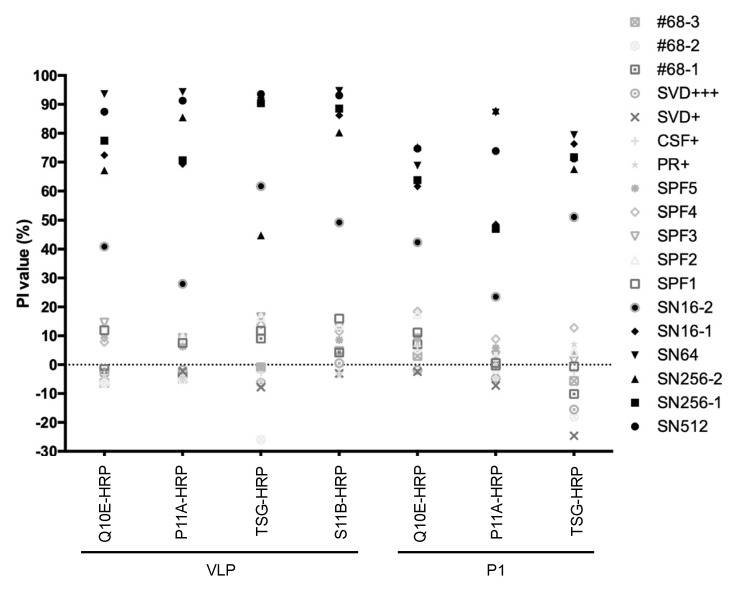
Specificity of VLP- and P1-based blocking ELISA paired with four MAbs. Samples included strongly positive anti-FMDV serum (SN512, SN256-1, SN256-2, and SN64), weakly positive serum (SN16-1 and SN16-2), SPF serum (SPF1, SPF2, SPF3, SPF4, and SPF5), anti-pseudorabies virus serum (PR+), anti-classic swine fever virus serum (CSF+), anti-swine vesicular disease virus serum (SVD+ and SVD+++), and FMDV-free samples (#68-1, #68-2, and #68-3). Each test was performed in triplicate.

**Figure 3 ijms-23-08542-f003:**
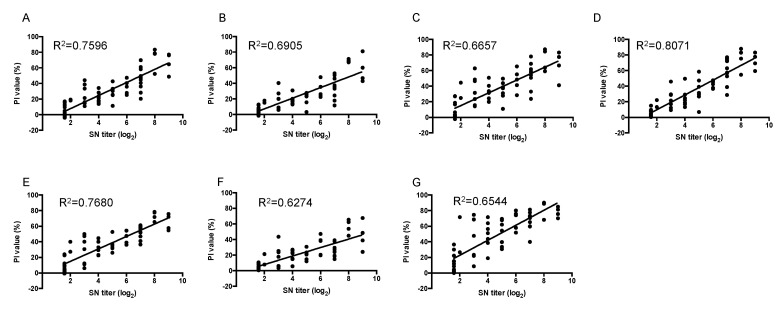
Scatter plots and calculated regression lines for SN titers and PI values as measured with solid-phase blocking ELISA based on VLPs (**A**–**D**) or GST-P1 (**E**–**G**). Results of VLP-based bELISA with tracers: (**A**) Q10EHRP, (**B**) P11A-HRP, (**C**) TSG-HRP, and (**D**) S11B-HRP. Results of P1-based bELISA with tracers: (**E**) Q10E-HRP, (**F**) P11A-HRP, and (**G**) TSG-HRP. Each test was performed in triplicate.

**Figure 4 ijms-23-08542-f004:**
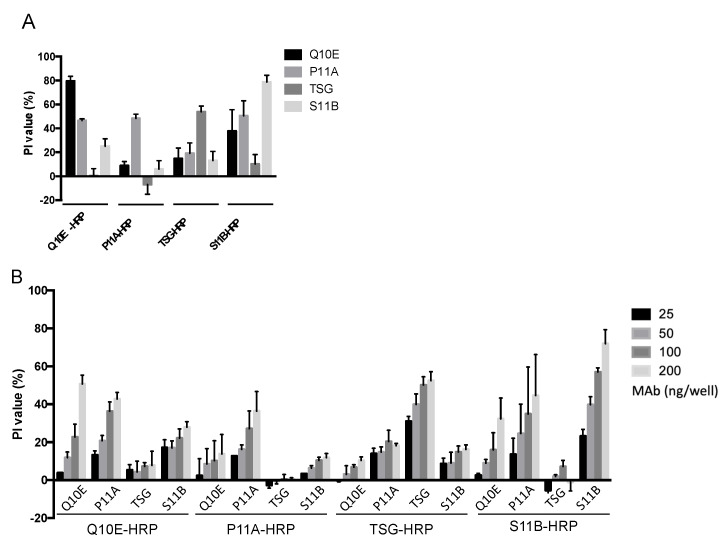
Steric effects between the four monoclonal antibodies (Q10E, P11A, TSG, and S11B). (**A**) The coated VLPs were blocked with non-conjugated monoclonal antibodies, including Q10E, P11A, TSG, and S11B at 400 ng per well, followed by 100 ng of the indicated conjugate. (**B**) The VLPs were blocked with different non-conjugated antibodies (25–200 ng per well), followed by 100 ng of conjugates. Each test was performed in triplicate.

**Table 1 ijms-23-08542-t001:** Characterization of four anti-FMDV capsid monoclonal antibodies.

	Q10E	P11A	S11B	TSG
Isotype	IgG2a/κ	IgG3/κ	IgA/κ	IgG1/κ
Unprocessed protomer	+	+	−	+
Neutralizing activity	+	+	+	−
Binding site	Site 1 (VP1-148)	Site 2 (VP2-72)	Site 3 (VP1-43,44)	Unknown

**Table 2 ijms-23-08542-t002:** Comparison of the performance of bELISA with different tracers and antigens.

Antigen		VLP								GST-P1					
Tracer		Q10E		P11A		TSG		S11B		Q10E		P11A		TSG	
SN	Number	+	−	+	−	+	−	+	−	+	−	+	−	+	−
<16	26	1	25	1	25	1	25	1	25	1	25	1	25	1	25
16–32	14	1	13	1	13	2	12	4	10	1	13	2	12	0	14
>32	23	17	6	17	6	17	6	22	1	18	5	15	8	14	9
Cut-off value		39.55		30.06		49.65		33.06		47.45		26.13		72.15	
Sensitivity *		73.91		73.91		73.91		95.65		78.26		65.22		60.87	
Specificity *		96.15		96.15		96.15		96.15		96.15		96.15		96.15	
Kappa value		0.71		0.71		0.71		0.92		0.75		0.63		0.58	

* Samples for SN16–32 were excluded from calculations of sensitivity and specificity.

## Data Availability

The data presented in this study are available on request from the corresponding author.

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
