# Peer review of "The Use of Distinctive Monoclonal Antibodies in FMD VLP- and P1-Based Blocking ELISA for the Seromonitoring of Vaccinated Swine"

_ijms, 2022, doi:10.3390/ijms23158542_

Round 1

Reviewer 1 Report

Heng-Wei Lee investigated in the present study entitled ‘The use of distinctive monoclonal antibodies in FMD VLP- and P1-based blocking ELISA for the seromonitoring of vaccinated swine’, to develop a blocking ELISA (bELISA) system based on VLPs or P1 combined with different tracer MAbs and evaluate the most suitable pairs.  For this purpose, established bELISA systems with VLPs and P1 and paired them with Q10E, P11A, S11B, and the non-neu-tralizing TSG MAb as tracers. Results showed two bELISA pairs, VLP:S11B-HRP and P1:Q10E-HRP, showed the highest performance among the seven and could be applied to large-scale serological testing.

The main strength of this original research article is that it addresses a timely question, evaluating the performances of different MAbs as tracers for the seromonitoring of vaccinated swine. In general, I think the results of this article are interesting and the authors’ fascinating observations on this topic may be of interest to the readers of the International Journal of Molecular Sciences. My overall judgment is to publish this article after some carefully revise.

Author Response

Thanks for your review.

After a thorough examination, we indeed found a sentence that was accidentally added in our manuscript in Materials and Methods in Line 375. We have removed it. Also, we amended Line 13, Line 107, Table 1, and Line 139. All “µl” was replaced by “µL” in Line 327, Line 336, and Line 364.

Reviewer 2 Report

In this manuscript, authors Lee et al. investigated and assessed the performance of blocking ELISA systems for utilization in seromonitoring the vaccine against the foot-and-mouth disease virus in cloven-hoofed animals. The blocking ELISA systems established were based on VLPs or P1 paired with different monoclonal antibodies used as tracers in the diagnostic system, and the performance of the various pairs were tested with numerous swine sera. The authors found that two bELISA pairs were positively correlated with serum neutralization titers and effectively detected protective antibodies. FMDV causes a serious and economically impactful infection in agriculturally important animals. This study focuses on improving a diagnostic system for seromonitoring swine vaccinated against FMDV. The experimental design is clearly described and scientifically sound. Overall, the manuscript is very well-written, and the results are clearly presented, with appropriate statistical analyses applied to the data. My only suggestion is that the authors indicate that the acronym OIE stands for the World Organisation for Animal Health for those readers who may not be familiar with the abbreviation.

Author Response

We appreciated your review, and we accepted your suggestion for showing the full name of OIE. Therefore, we added the information in Line 48 where “OIE” is displayed in the first place.

The virus strain, O/TW/97 (O/97), was characterized by porcinophilic activity [3]. Following the guidelines of the OIE (Office International des Epizooties/World Organisation for Animal Health), mandatory vaccination with inactivated vaccines made using three strains—O 4174, O1 Campos, and O1 Manisa—was recommended by the World Reference Laboratory based on an evaluation of r1 values greater than 0.3 [4] immediately following the outbreak.